# The Syngameon Enigma

**DOI:** 10.3390/plants11070895

**Published:** 2022-03-28

**Authors:** Ryan Buck, Lluvia Flores-Rentería

**Affiliations:** Department of Biology, San Diego State University, San Diego, CA 92182, USA; lfloresrenteria@sdsu.edu

**Keywords:** syngameon, hybridization, multispecies

## Abstract

Despite their evolutionary relevance, multispecies networks or syngameons are rarely reported in the literature. Discovering how syngameons form and how they are maintained can give insight into processes such as adaptive radiations, island colonizations, and the creation of new hybrid lineages. Understanding these complex hybridization networks is even more pressing with anthropogenic climate change, as syngameons may have unique synergistic properties that will allow participating species to persist. The formation of a syngameon is not insurmountable, as several ways for a syngameon to form have been proposed, depending mostly on the magnitude and frequency of gene flow events, as well as the relatedness of its participants. Episodic hybridization with small amounts of introgression may keep syngameons stable and protect their participants from any detrimental effects of gene flow. As genomic sequencing becomes cheaper and more species are included in studies, the number of known syngameons is expected to increase. Syngameons must be considered in conservation efforts as the extinction of one participating species may have detrimental effects on the survival of all other species in the network.


*Nearly all organisms, met with in nature as well as under cultivation, man included, are hybrids which were mistakenly considered to be specifically pure, so that their behaviour was unconsciously held to be that of specifically pure organisms, while it was that of hybrids; so it happened that segregation was mistaken for heredity.*
—Lotsy 1916

## 1. Introduction

Interspecies hybridization is relatively common across taxa, with occurrence estimates of 25% in plants and 10% in animals [1], and is thought to be the cause of several major speciation events [2,3]. When a group of otherwise distinct species are connected by hybridization, they form a syngameon, a copulative community [4]. The discovery of the first natural syngameon in birch trees by Gunnarsson and the multidirectional hybridization of cultivated *Saxifraga* by Lloyd (in [4]) sparked the first serious investigations into hybridization itself, which continue to this day. From its inception, the word syngameon was used to collectively describe “a large number of different individuals [from different species], which are all apparently able to produce fertile offspring with one another; one very large pairing-community, one syngameon” [4]. However, shortly thereafter, Du Rietz [5] used the term to describe a polymorphic hybrid collective in which “species have got more or less lost”, precipitating an idea that syngameons were just a collection of taxonomic misfits unable to be classified. Later, other authors such as Cuenot [6], Grant [7], and Beaudry [8] reclaimed the word to mean “species linked by frequent or occasional hybridization, a hybridizing group of species”. Out of this definition was born the ecological species concept, which allowed for gene flow but separated species by their adaptations to particular niches, in an attempt to explain oak differentiation under gene flow. The now well-known oak syngameon was alternatively named a “multispecies” by Van Valen [9]. Templeton [10] began a pattern of syngameon misinterpretation when he conflated the term with a hybridizing pair, the aftermath of which can be seen in recent hybridization papers that use syngameon to describe any two species that hybridize (e.g., [11,12,13,14]. According to Boecklen [15], a syngameon is produced when a group of closely related species forms a complex set of hybrid combinations. We recommend this use of syngameon to define a breeding system with a three or more multispecies network. The minimum number of three participating species is important in this context as it distinguishes the commonly studied hybridizing species pairs from a more complex and possibly synergistic interacting multispecies system [16]. It is important to note that the current definition does not distinguish between fertile and sterile hybrids, or diploid and polyploid systems, both of which could have varying impacts on the structure of the syngameon. However, the term seems to be restricted to naturally occurring hybrids, whereas the term coenospecies would refer to artificial hybrids (Glossary) [17].

Despite the misused language, the amount of described syngameons remains extremely low in comparison to the number of hybridizing species pairs and thus far is generally restricted to plant taxa (Table 1). Only recently have researchers begun to hypothesize how these rare complexes are able to overcome numerous reproductive barriers in the process of their formation [18,19,20,21,22,23,24,25,26]. Even less is known about how these interactions are maintained over time or if they are perpetually unstable. Additionally, the evolutionary consequences of sustained multispecies gene flow remain unexplored, leaving the future of syngameons speculative at best. In this review, we will explore three questions: (1) how do syngameons form, (2) how are they maintained over evolutionary time, and (3) why are they so rare? Lastly, we will discuss the future of syngameons in light of a changing world and provide some recommendations relevant to conservation.

## 2. How Do Syngameons Form and Collapse?

### 2.1. The Origin of Syngameons

For hybridization to occur, species must overcome any existing barriers to gene flow, which include pre- and postzygotic reproductive isolating mechanisms. Prezygotic barriers can consist of temporal, geographic, mechanical, behavioral, and genetic mechanisms, while postzygotic barriers can consist of hybrid sterility, hybrid inviability, and F_2_ breakdown [145]. While not always initially present, some of these barriers can form after the secondary contact of two lineages to prevent further hybridization and reinforce species boundaries [146]. Conversely, the initial lack of reproductive isolating mechanisms or the failure of reinforcement (Glossary) can lead to stable hybrid zones. Despite the numerous obstacles faced, hybridization is not rare [1,147]. So why, then, are syngameons so rarely reported? After all, syngameons are just hybridization events between three or more species. What makes adding this third species interaction so difficult? The answer may lie in how syngameons form and collapse.

### 2.2. The Birth and Death Hypotheses

#### 2.2.1. The Rapid Radiation Hypothesis

The rapid radiation hypothesis [21,148] postulates that rapid radiations, or in other words, relatively quick and numerous speciation events, allowed for the repeated origins of hybrid lineages. Syngameons are able to form among the newly radiated species because reproductive isolating mechanisms have yet to develop (Figure 1). In turn, this gene flow can act as a catalyst for additional radiation by replenishing standing genetic variation, aiding in the consumption of unexploited resources and occupation of new niches [149]. Further, these hybrid lineages could speciate themselves, becoming hybrid species; however, this often requires the formation of reproductive barriers, which would exclude the newly formed species from the syngameon. Seehausen [148] used Heliconius butterflies to exemplify syngameons providing new adaptive traits and promoting ecological diversification. Using kiwifruit as an example, Liu [21] showed how syngameons developed during early radiation but later collapsed as species diversified into new ecological opportunities to reduce contact and competition (but see, ref. [150]). The classic examples of radiations, including Heliconius butterflies [151,152], Darwin’s finches [153,154], and African cichlid fish [155], showed a similar pattern as species numbers rose and underutilized resources became scarce, stabilizing selection occurred and species began to accumulate genomic incompatibilities [148]. While the concepts behind the hypothesis remain valid, it is difficult to prove if ancient syngameons formed during radiation events, as many participating species may have since gone extinct and the rapid timeline of diversification would make a transient hybridization event hard to detect. Current simplified methods to detect ancient hybridization (e.g., ABBA-BABA) can fail to distinguish population structure from actual introgression when population sizes are large [156], as could happen in syngameons and rapid radiations. However, with improving molecular and coalescent techniques, ghost lineages and ancient introgression events are becoming easier to trace [157,158,159].

#### 2.2.2. Surfing Syngameon Hypothesis

Rather than rapid radiation events causing syngameons, the surfing syngameon hypothesis [14,160] suggests syngameons that form during island colonization events can both cause and prevent rapid radiations. Distinct colonizing genotypes (referred to by Caujapé-Castells as morpho-species or incipient species) that were previously isolated on the mainland but are phylogenetically close enough to have gene flow can form a syngameon during a colonizing event of a new island (Figure 2A,B). The increase in genetic diversity would be enough to overcome selective pressures and founder effects, promoting the colonization of syngameon participants. Using species in the Canarian archipelago, Caujapé-Castells [160] indicated that this type of event could be detected through the level of endemic species, with low levels of endemism resulting from syngameon colonization and high levels from the formation of incompatibility barriers. In low-complexity islands, syngameons would stall evolutionary change due to high levels of gene flow homogenizing genomes, thus preventing rapid diversification. In high-complexity islands, syngameons would promote adaptations due to the high genetic diversity hybridization provides; therefore, resulting in rapid radiations and the eventual collapse of the syngameon (Figure 2C) due to competition and the formation of reproductive barriers [14]. Future studies should test the validity of this hypothesis beyond the Canary Islands and examine if it is broadly applicable to other allopatric dispersal scenarios such as the colonization of nearby niches or mosaic hybrid zones.

#### 2.2.3. Edge Range Hypothesis

Syngameons may form at the edges of species’ ranges, where multiple species can overlap in distribution (Figure 3A) [23]. Typically, range edges are seen as population sinks because the species is unable to adapt to the new, local environment beyond the current distribution boundary [161]. However, hybridization at a species’ range edge may facilitate survival by introducing locally adapted or novel traits through introgression [161]. Cronk and Suarez-Gonzalez [23] used a poplar syngameon to show how a tri-species interaction allows for the increased survival of hybrids at the edge of species boundaries. They also illustrated how as ranges expand and contract, these gene flow events could be episodic, explaining patterns of ancient introgression followed by divergence, then introgression again. However, Ottenburghs [162] pointed out that these “merging-and-diverging cycles” could result in the build-up of genetic divergence during allopatric phases, leading to lower levels of introgression during the following sympatric phase, eventually ending with a collapse of the syngameon. Additionally, Cronk and Suarez-Gonzalez [23] failed to consider the stability of syngameons at range edges because these interactions could lead to the formation of species barriers and thus the collapse of the syngameon, or even the formation of a new hybrid species with higher fitness than its parental species. Hybrid speciation could lead to the collapse of the syngameon and the possible extinction of the parent species via genetic swamping (Glossary) or hybrid superiority [163]. Moreover, there are examples of syngameons that do not form at the range edges, such as in Quercus in which some species overlap in wide ranges of the distribution [23]. As studies expand their scope beyond hybrid pairs to include more hybridizing species, range overlaps should be closely investigated in order to revisit this hypothesis under more scrutiny.

#### 2.2.4. Genomic Mutualist Hypothesis

Lastly, Cannon and Lerdau [18] hypothesized that species form syngameons by acting as genomic mutualists. In their scenario, multiple species remain partially interfertile with each other but experience divergent selection on portions of their genome, while low levels of neutral or adaptive gene flow occur in other parts of the genome. This creates a balance between purifying selection within species for specific phenotypes and diversifying selection among species for novel phenotypes. To avoid the negative consequences of extensive gene flow, species would develop a reduced but persistent capacity for interspecific mating, making periods of gene flow infrequent, episodic, and often unidirectional; however, in some systems, syngameons are multidirectional and often reciprocal gene flow occurs in different magnitudes [23,119,123]. These mating interactions are largely controlled by the quantity and quality of pollen or sperm, so interspecific gene flow would often be triggered by the decline of one species (Figure 4A,B), resulting in an overabundance of heterospecific gamete landing on the rare species (Figure 4B,C). This in turn allows for the rare species to avoid local extinction and inbreeding depression through the maintenance of diversity, a process called genetic rescue (Glossary; Figure 4A–C) [164,165]. However, demographic swamping (Glossary; Figure 4D), or genetic swamping (Figure 4E), where rare species are replaced by hybrids [163], is often used to counter this hypothesis as too much gamete swamping could instead result in the proliferation of hybrids and extinction of the rare species. Cannon and Scher [20] suggested that Mendelian segregation and pollen competition allow for the formation of genetic bridges among species and thus the participation in syngameons. They argue that small proportions of the gametophytes produced by F_1_ hybrids would be 80–90% identical to a gametophyte produced by one of the parental species. That small portion (which could total millions of gametophytes in a heavily producing system such as oaks) coupled with conspecific pollen advantage, could result in a backcross generation nearly identical to the parental types, making introgression possible without the erosion of genetic coherence. Although they used simulations based on a real oak syngameon, they limit their hypothesis to organisms with low chromosome numbers, copious gamete production, conserved genomic structure, and conspecific gamete advantage. More syngameons are being uncovered that do not follow these strict assumptions, thus future simulation studies will need to broaden their scopes and reassess the genomic mutualist hypothesis.

### 2.3. Spatial Limitations

With the various ways syngameons are thought to form, it seems that there should be an abundance of syngameons. Perhaps the limiting factor is species distribution, meaning that, despite the number of species pairs with overlapping distributions reported to hybridize, the chances to have multiple hybridizable species with overlapping distributions is limited. In describing competition among highly diverse tropical tree communities, Cannon and Lerdau [18,25] found that direct spatial proximity with close relatives was infrequent, so even in complex ecological landscapes, the chances of overlapping with a congeneric species is low. Yet even if direct spatial overlap does not frequently occur, pollen and seed could still disperse into adjacent habitats and trigger syngameonic behavior. If, however, sympatry does occur, species in a syngameon could coexist and avoid competition by diversifying into microhabitats, as demonstrated by Schmitt et al. [137] in a Neotropical syngameon. Similarly, differing patterns of speciation may also play a role in limiting syngameon formation because allopatric species coming into secondary contact could be less likely to share a large enough portion of their range to overlap with more than one species. Further, the narrow hybrid zones that can result from secondary contact do not allow for the introgression of genes beyond the hybrid zone itself, which is usually at the edge of species’ ranges. Alternatively, F₁s could form but reproductive barriers could prevent any backcrossing with the parental species, thus introgression would not occur, as seen in Ligularia [110]. While many syngameon participants defined here hybridize with the same single species, genes are not necessarily introgressed across all species’ ranges, especially if the species hybridize at opposite ends of a range (Figure 3B,C). While this would still technically be considered a syngameon, the participants are not receiving all the benefits of the network-like structure of more sympatric syngameons. Cases of sympatric speciation may create more opportunities for multiple species to have overlapping distributions; however, these scenarios are rarer [166,167] and usually result in the formation of a reproductive isolating barrier [162,168,169], which would likely prevent any further hybridization. While the above syngameon formation hypotheses are not necessarily mutually exclusive, maintaining hybridization in multiple species at once can have compounding complications, with genetic swamping, lineage collapse, and the formation of reproductive barriers, all challenging the stability of a syngameon. If syngameons constantly fight to exist, then how are they maintained over time? The structure of known syngameons may shed light on this perplexing question.

## 3. How Are Syngameons Maintained over Evolutionary Time?

Most of the formation hypotheses above mention the episodic occurrences of gene flow within a syngameon and the limited amount of gene flow that must occur to stabilize the interactions. Yet most known examples of syngameons show extensive and constant gene flow among numerous participants (Table 1). This discrepancy in theory and practice may be due to the varying hubs of introgression (Glossary), where some species contribute more genetic information than they receive and are connected to a large number of other species through gene flow [28]. The number of participating species can vary over geographical space and evolutionary time, with a single species (referred to as a hub) that has direct contact with multiple species, and as a result, genes passively introgress through the various pathways radiating from the hub (Figure 5). In these hub-based networks, if one pathway collapses, gene flow can still be maintained through the numerous other pathways connecting the species together, as long as there are no geographic or intrinsic barriers that act to contain alleles to one hybridizing species pair. However, if a hub disappears, that will likely have a larger effect on the entire network, but the extent of that effect is not currently known.

The direction and magnitude of these introgression pathways are rarely uniform. Boecklen [15] used simulations to test the structure of nine natural and four artificial syngameons (Glossary), finding that a majority exhibit a nonrandom structure, with a few species dominating the patterns of introgression. He concluded that geographically widespread species would have more opportunities to hybridize than restricted ones, with the *Boechera* syngameon demonstrating a positive relationship between geographic range and the number of mating combinations. The same is seen in North American white oaks, with the widespread *Quercus alba* mating with 11 out of 14 species in the syngameon [23]. It seems that the distribution of a species has a large impact on its ability to participate in the syngameon, with large contributors maintaining the structure of syngameons across geographical space. However, geographically widespread species may encounter more geographic and ecological barriers that could lead to population structuring or barriers to gene flow [170]. These could ultimately prevent the species, or at least certain populations, from participating in the syngameon or could result in reduced introgression beyond the hybrid zone. Additionally, the propensity to hybridize was unequal, even when species had equal opportunities to hybridize [15]. This suggests that there are other factors beyond range that affect the direction and magnitude of introgression within a syngameon. Genetic distance (Glossary) may be the largest of these factors, with closely related species hybridizing more readily than distant ones [15,171]. This would mean that the structure of syngameons is partially dependent on the relatedness of the species participating. Hypothetically, as time passes, species would become more distinct, compromising the structure and putting the maintenance of the syngameon at risk. However, the occasional gene flow events among syngameon members would counteract divergence and keep genomic distance smaller.

There are several cases (e.g., coral [28] and pinyon pines [120]) where gene flow can favor one direction within a syngameon. The reasons for unidirectionality are numerous but include postzygotic barriers that prevent one parent from backcrossing with the hybrid offspring, such as hybrid inviability, hybrid sterility, and F₂ breakdown [150,172,173,174]. This could promote a stable syngameon by preventing maladaptive hybrid derivatives from forming and only allowing the viable and fertile backcrossed individuals to proliferate. In this sense, the formation of reproductive barriers can actually maintain syngameons rather than collapsing them by preventing hybridization. On a genetic level, the uneven exchange rate of loci may represent regions that maintain functional differences between species [28]. In corals, large sections of non-introgressing genes were found among species with high levels of overall gene flow [28]. This suggests that loci responsible for differentiating species may be linked to loci that contribute to reproductive isolation, creating gene regions that maintain individual lineages in a syngameon, while still allowing for some gene flow. In hybridizing species of *Drosophila*, recombination rates may be reduced while chromosomal inversion rates are increased to promote divergence under gene flow, yet maintain high diversity in the rest of the genome [175]. However, selection could reduce diversity in genomic regions and result in a similar, but misleading pattern as non-introgressing loci [176]. Future studies will need to take both mechanisms into account by examining diversity across the whole genome, especially when taxa have recently diverged [176].

The evolutionary advantages and disadvantages of interspecific gene flow are well understood [3,145,177,178], but it is not known if these consequences are the same in these multispecies networks. Cannon and Petit [16] suggest that syngameons have synergistic properties, with network-like benefits that total more than just the sum of individual species pairs. Schmitt et al. [93,137] suggest that two contrasting evolutionary pressures are constantly acting on a syngameon, one at the species level to maximize individual species’ fitness and reduce competition among species, and one at the syngameon level to increase genus survival and maximize population size. In syngameons, adaptive introgression can maintain hybrid zones through the sharing of beneficial alleles [24]. Natural selection plays a role in maintaining the poplar syngameon when adaptive alleles are episodically exchanged across species boundaries [123]. Syngameons can also have increased heterozygosity, while maintaining partial infertility among species [77]. In the Fabaceae family, this partial infertility prevents genomes from fully merging, while still allowing gene flow to increase heterozygosity [77]. Levi et al. [26] suggested syngameons could help fuel the Red Queen arms race (Glossary) in tropical trees by increasing heterozygosity and introducing novel phenotypes. These beneficial outcomes of gene flow help maintain syngameons and can counter the negative complications that arise with hybridization. While the current definition does not differentiate between fertile hybrids that can backcross with their parental species and sterile hybrids that would prevent introgression, the hypothesized synergistic effects would likely only exist in the former situation where adaptive traits can be passed across species barriers. Further, the creation of infertile hybrids would more likely result in demographic swamping (Glossary; Figure 4D) and be detrimental to the syngameon as a whole.

There are many ways that syngameons can remain stable over long periods of evolutionary time including uneven participation, geographic distribution, genetic distance, and direction of gene flow within a syngameon. These factors can allow gene flow to occur episodically or in minute amounts, preserving the beneficial aspects of hybridization while avoiding the detrimental ones. A common misconception with hybridization is that it is ephemeral and only a stopping point on the way to reproductive isolation [179]. While time since divergence is positively correlated with the strength of reproductive barriers [180], classic two-species hybrid zones can be stable over evolutionary time through the balance between selection and dispersal [181], so it is reasonable that multispecies hybrid zones, while more complex, can be maintained in the same way. Without strong selection for the formation of reproduction barriers and with occasional gene flow partially homogenizing genomes, isolating barriers may take even longer to form within a syngameon, if at all. Cannon and Petit [16] argue that syngameons do not have to be transitional or incipient phases on the way to complete speciation because reproductive isolation is not a requirement for speciation in the first place. We assert that while syngameons can be ephemeral and collapse if reproductive barriers form, they can also last for as long as species themselves, constantly fluctuating and evolving. With the potential stability of known syngameons over time, why are we just now discovering syngameons and why have we not detected more? Both the past and future of science hold the answer.

## 4. Why Are Syngameons So Rare?

Hybridization in general was overlooked for decades. Considered infrequent and not important to evolution, it remained unexplored for the better half of the 1900s. Although extensive efforts were eventually made, studies of hybridization were limited to phenotypic comparisons [182]. This initial lack of genomic data could be the main reason so few syngameons have been uncovered. With the incorporation of next-generation sequencing (NGS) and whole-genome data, more hybridization events are being discovered, and thus more syngameons are being uncovered (Table 1, Figure 6). Likewise, scientists are starting to recognize the importance of hybridization events and are able to describe patterns of reticulated evolution, so it is only a matter of time before more syngameons are reported.

While our overall detection methods are improving, several factors can still prevent the discovery of syngameons. Cryptic hybrids, which are genetically of hybrid origin but morphologically appear identical to one parental species, are one such preventing factor. Ladner [28] found cryptic hybrids in corals and Buck et al. [119] found cryptic hybrids in pinyon pines, both cases exemplifying the issue that syngameons cannot be detected if hybrid individuals are not known to exist. It makes one wonder how many other systems have individuals of hybrid origin hidden among their parental species. The increased use of a combined morphological and genetic approach should help reveal cryptic hybrids in the future. Unexplored hybrid pathways are another limit on syngameon detection. Most studies explore hybrid zones by looking at two parental species and their resulting offspring without considering the potential for multispecies introgression. As studies expand to incorporate more species, we may find that hybridization extends beyond species pairs into syngameons. In pinyon pines, for example, Buck et al. [119] found that a complex originally thought to be composed of two hybridizing species actually consisted of three species undergoing tridirectional gene flow. The same pattern was found in poplars [123]. The genetic bridge hypothesis [20] postulates another reason why syngameons may go undetected. The minute amounts of genetic information that are introgressed from an F_1_ hybrid back into a parent species through backcrossing may result in gametes that are indistinguishable from the parental species. This is similar to the cryptic hybrids problem, except that the genetic bridge between species is undetectable, while the hybrids rarely pass the backcrossed F_1_ generation. Finally, as brought up in the rapid radiation hypothesis, it is difficult to prove that syngameons occurred in the past so they may remain hidden by time until molecular, coalescent, and ancient introgression techniques improve [162].

While the potential for future syngameon discovery could increase with the incorporation of new technology and more species, the global climate is rapidly changing due to anthropogenic activities [183]. What does this mean for the future of syngameons? Can syngameons generate the right combination of genes to save the member species from extinction? How will climate change affect syngameons and how can we conserve species that participate in gene flow networks?

## 5. The Conservation and Future of Syngameons

With climate change, several species ranges are shifting polewards or disappearing altogether [184,185] and novel interspecies interactions are being established [186]. These migrations and novel interactions could lead to new hybridization events between previously isolated species [162] and could result in the formation of syngameons, especially at range edges [160] and during colonization events [23]. Additionally, anthropogenic introductions and disturbed habitats can create novel niches and allow hybrids to thrive [162,187]. The incorporation of adaptive alleles, heterozygosity, and an increase in effective population size via participation in a syngameon could be critical to the survival of species in a quickly changing climate [16,18]. Conversely, contracting ranges and increasing extinction rates could result in the collapse of syngameons if participating species begin to disappear or become allopatric. However, the degree to which one species affects a syngameon as a whole remains unknown.

The focal unit of conservation is a species. The definition of what constitutes a species is widely debated [188]. A “species” under the Endangered Species Act [189] (but see 1978 amendment) includes “any subspecies of fish, wildlife, or plants and any other group of fish or wildlife of the same species or smaller taxa in a common spatial arrangement that interbreed when mature”. However, this loose definition does not take hybridization into consideration. Gene flow has always been a taxonomic issue since the early species concept debates. Hybridization, especially at a multispecies level, went against the standing concept of a biological species [190]. Thus, the discovery of syngameons made taxonomists rethink species concepts, leading [68], who studied an oak syngameon, to create the ecological species concept. However, arbitrary cutoffs must be made to distinguish niches and some syngameon participants may not occupy different niches. Therefore, it is more appropriate to consider the whole syngameon as a biological conservation unit [23]. While individual members of a syngameon are not reproductively isolated from each other, syngameons are isolated from other syngameons and non-participating species [187]. However, conservation efforts should not necessarily treat syngameons as they would a single species, because doing so would essentially collapse all the lineages into one and reduce the conservation importance of the individual species. Each participating species should be conserved with the assumption that individual contributions have widespread effects across the whole multispecies network [179]. This is particularly important as it has been suggested that in some instances this multidirectional and recurrent hybridization has created new hybrid species [86,120,191,192,193]).

With limited funds, conservationists often find they cannot protect every species but must focus their efforts. In the case of a syngameon, the structure must be taken into account, with a priority on hub species which have a larger effect on the network as a whole. As many hub species encompass larger ranges, they do not usually represent a conservation concern, and protecting them might require a large amount of resources. However, we argue they need to be considered in conservation genetic plans as they harbor important genetic diversity needed for the evolution of the complex. As an individual’s effects on the structure of a syngameon are still largely unknown, it is hard to predict how the loss of one species could affect the syngameon as a whole. The decline of a single species may result in the collapse of the whole syngameon and potentially lead to the extinction of the remaining species. Alternatively, if one population is participating in the syngameon but the rest are not [35], as is possible in edge-range syngameons, limited conservation efforts can equally focus on that population and core populations to preserve syngameon structure. Lastly, gene flow should be considered as a potential tool for conservation because the immediate increase in heterozygosity and the adaptive introgression of beneficial alleles could be critical to the survival of endangered species, with some authors arguing that the benefits outweigh the potential dangers [194,195,196]. However, human-induced hybridization events should be carefully planned and controlled to avoid the outbreeding depression effects seen in unregulated anthropogenic gene flow events [162]. This highlights the need for a systematic change in the legal framework of conservation policy. Current conservation efforts are reserved for well-defined species, while hybrids are largely ignored and discounted as “genetic erosion” or “pollution” [162]. The Endangered Species Act should expand its protection to not only hybridizing pairs but also complexes like syngameons.

## 6. Conclusions

Almost a century ago, Lotsy [4] recognized the complexity of a syngameon as species that readily mate among several species but also recognized the difficulty of detecting multidirectionally hybridization by his statement, “Can a careful study in nature… reveal the true relationship between the various individuals within the genus, can it decide which of the forms are hybrids, which species and from which combination of the latter the hybrids arose? To my way of thinking, not”. The advancement of next-generation sequencing has opened the possibility to carefully and precisely answer these questions. Not only has this technology enabled us to detect multidirectional hybridization, the magnitude of gene flow, and the percentage of the parental ancestry, but it has also demonstrated that syngameons are not as rare as previously thought. In our comprehensive literature review, we found that over the past century, reports of syngameons have increased in relation to the use of genetic markers. Just over the past decade, the numbers have increased (Figure 6) and we predict they may keep rising. Future syngameon studies should focus on understanding how syngameons form and remain stable over long periods of evolutionary time. As more syngameons are discovered, formation hypotheses can be tested and compared. Combining biogeographic and population-level genetic data may give insight into ancient introgression events that coincide with range contacts, colonizations, and rapid radiations. More simulations can be run to detect the structure of syngameons, which may shed light on individual species’ roles in these multispecies networks. Discovering how the individual species affects the structure of a syngameon as a whole remains the largest enigma of the syngameon. If ranges contract out of sympatry or species go extinct, researchers can examine the resulting effects on the other participating species.

## Figures and Tables

**Figure 1 plants-11-00895-f001:**
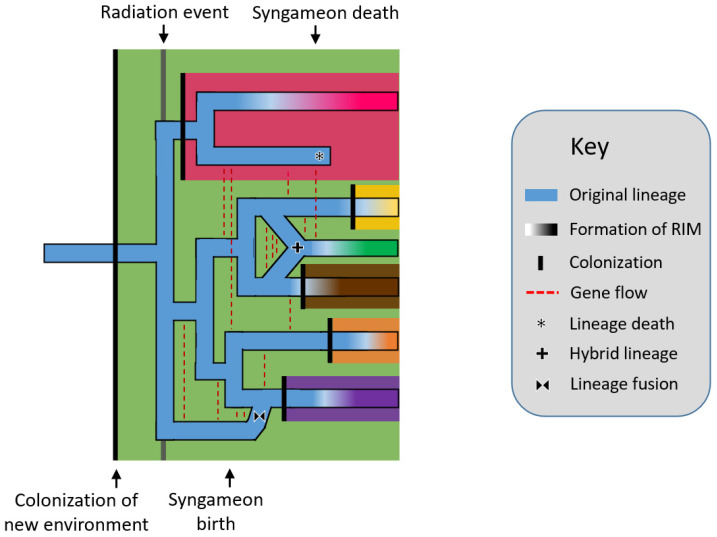
Rapid radiation hypothesis showing a lineage (blue horizontal line) colonizing a new environment (black vertical line), which eventually triggers a rapid radiation event. Speciation is followed by gene flow events (red dashed lines) which form a syngameon. The eventual collapse of the syngameon occurs when reproductive isolating barriers form among species, usually after the colonization of new environments, leaving two or no species with interspecific gene flow. Several potential outcomes are shown including hybrid speciation (plus symbol), extinction (asterisk), and fusion (bowtie symbol). RIM = reproductive isolation mechanism.

**Figure 2 plants-11-00895-f002:**
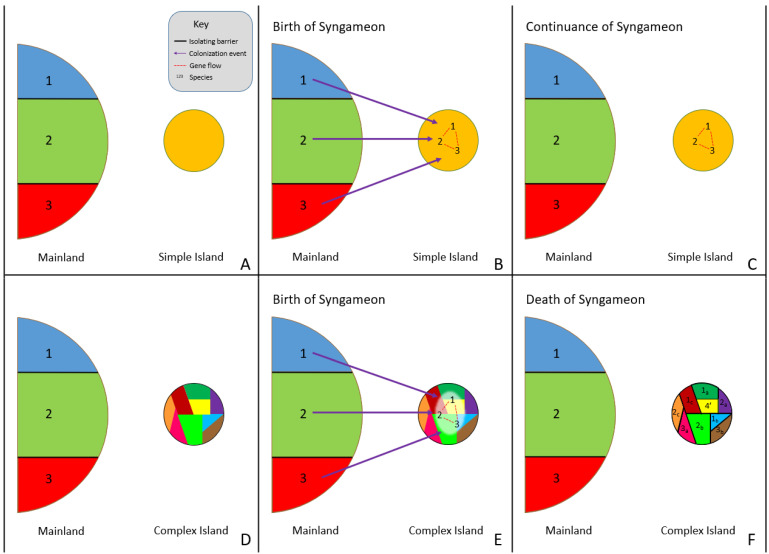
Surfing syngameon hypothesis, in which previously isolated species (1, 2, and 3) come into contact during the colonization of a low-complexity island (**B**) and high-complexity island (**E**), resulting in hybridization and the formation of a syngameon (**B**,**E**). The syngameon increases genetic diversity and reduces the effects of bottleneck events, resulting in the successful colonization of an island. If the island is open and uniform (**A**–**C**), with little to no ecological and geographical complexity (simple island), then evolutionary change is slowed down by syngameonic introgression/gene flow, resulting in homogenization of traits and the continuation of the syngameon (**C**). If the island is geographically and ecologically complex (**D**–**F**), then selection, adaptation, and competition eventually drive divergence and the formation of reproductive isolating barriers, resulting in the eventual collapse of the syngameon (**F**). Participation could even result in the creation of a new hybrid lineage (**F**, shown as lineage 4′).

**Figure 3 plants-11-00895-f003:**
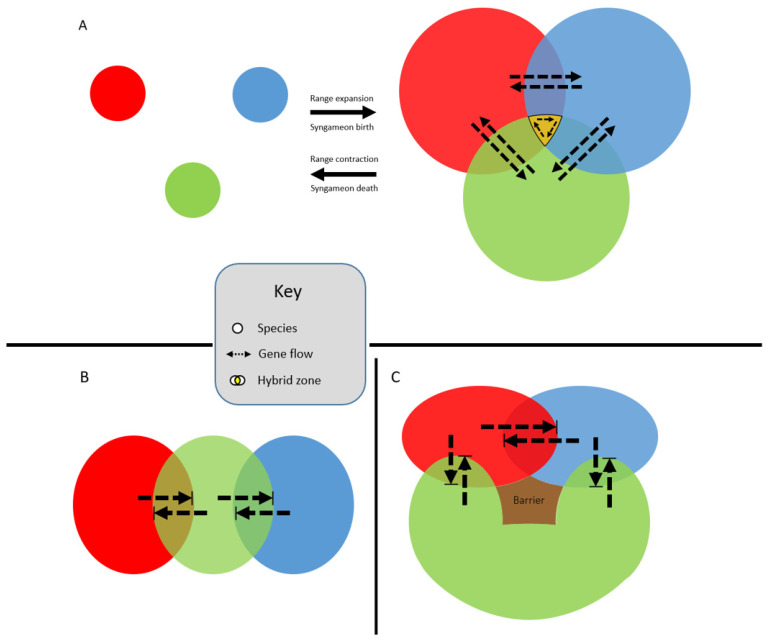
Edge-range hypothesis whereby the expansion and contraction of species’ ranges (**A**) over time makes gene flow within the syngameon episodic. This allows for the retention of species’ identities while still allowing for the exchange of adaptive alleles (dashed arrows). A caveat to the edge-range hypothesis is that all three species’ ranges rarely overlap (shown in gold). More probable scenarios are shown in (**B,C**), where species’ distributions overlap independently. While still technically syngameons, the scenarios represented in (**B**,**C**) may result in introgression not extending past the hybrid zones (bounded box), resulting in local admixture directly between hybrid pairs but no genes are shared indirectly through introgression via a third species.

**Figure 4 plants-11-00895-f004:**
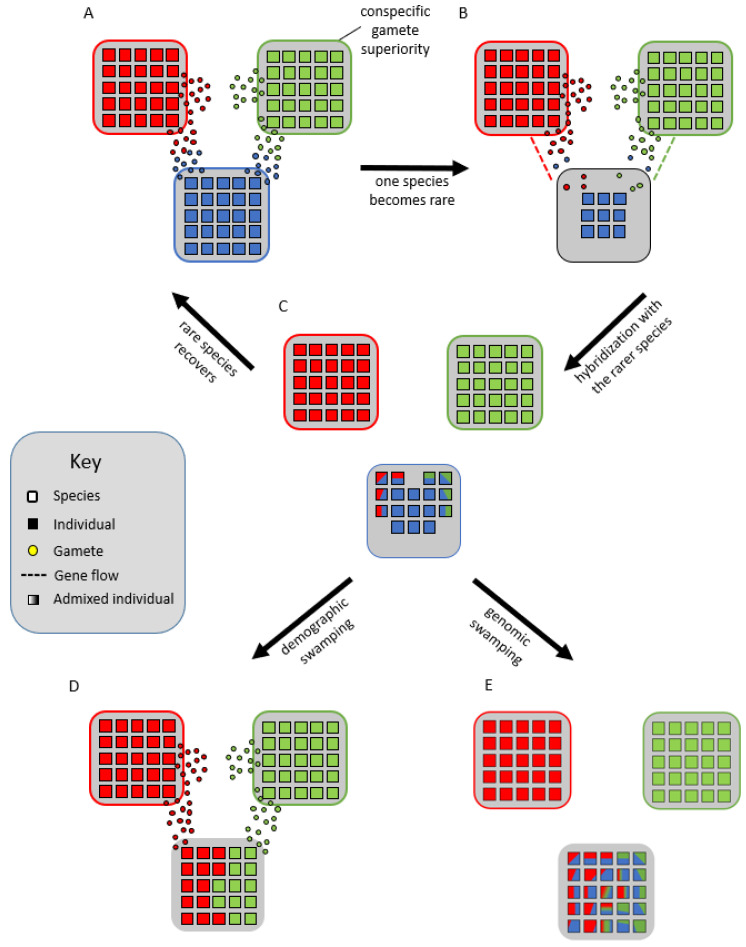
Genomic mutualist hypothesis in which there is a reproductive barrier favoring conspecific gametes (**A**) until one species becomes rare (**B**), wherein the gamete load from interspecific donors forces the rarer species to hybridize (**C**). This could lead to the rarer species benefiting from the increased genetic variation and effective population size, allowing it to overcome inbreeding depression and recover, a process known as genetic rescue (**A**). Alternatively, the rarity could lead to demographic swamping, where the rare species is replaced by the more abundant species through the purging of maladaptive hybrids (**D**), or genetic swamping, in which the rare species is replaced by admixed individuals (**E**).

**Figure 5 plants-11-00895-f005:**
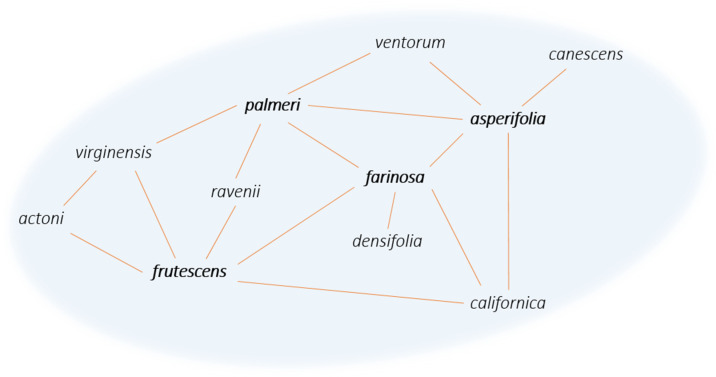
Network diagram of the *Encelia* syngameon (adapted from [92]). Lines connecting species represent gene flow, with species such as *E. frutescens*, *E. farinosa*, *E. palmeri,* and *E. asperifolia* exemplifying hubs of introgression (in bold).

**Figure 6 plants-11-00895-f006:**
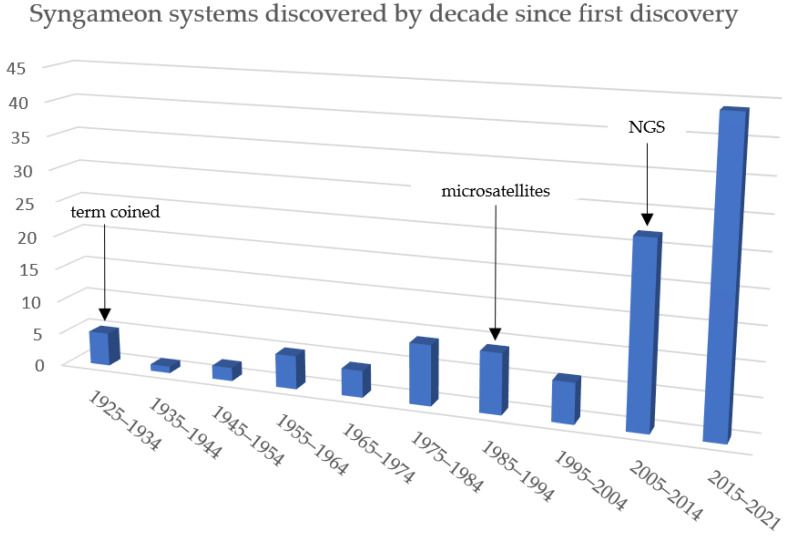
The number of syngameon systems discovered by decade (as of 21 December), showing an increase in the number of discoveries, especially in the 2010s.

**Table 1 plants-11-00895-t001:** Known Syngameons. List of genera with known syngameons and their common names in parentheses. Their number of participants and the taxonomic kingdom they belong to are listed in the following columns.

Genera (Common Name)	Known Participants	Kingdom	Source
*Acropora* (Coral)	8	Animalia	[27,28] *
*Anser* + *Branta* (Geese)	15	Animalia	[29]
Artibeus (bats)	3	Animalia	[30]
*Callithrix* (marmosets)	3	Animalia	[31]
*Canis*	3	Animalia	[32,33] ^2^
*Carabus* (Carabid beetles)	6	Animalia	[34]
*Catostomus* + *Chasmistes* + *Deltistes* (catostomid fish)	4	Animalia	[35] *
*Cerion* (snail)	Not specified	Animalia	[36] *
*Colias* (sulfur butterflies)	3	Animalia	[37] *
*Daphnia* (plankton)	5	Animalia	[38] (but see [39])
*Desmognathus* (Dusky Salamanders)	3	Animalia	[40]
*Drosophila*	At least 3	Animalia	[41,42]
*Eueides* (butterflies)	5	Animalia	[43]
*Geospiza* (Darwin’s finches)	Two sets of 3	Animalia	[44,45,46]
*Habronattus* (jumping spiders)	At least 3	Animalia	[47]
*Heliconius* (butterflies)	One set of 3; one set of 4; one set of 9	Animalia	[43]
*Homo*	3	Animalia	[31,48]
*Konia* + *Myaka* + *Pungu* + *Sarotherodon* (Cichlid fish)	8	Animalia	[49] *
*Liolaemus* (lizard)	4	Animalia	[50]
*Montastraea* (coral)	3	Animalia	[51] * (but see [52])
*Pacifigorgia* (octocorals)	Not specified	Animalia	[53,54] ^2^
*Papio* + *Theropithecus* (baboons)	at least 3	Animalia	[55] *
*Psammocora* (Indo-Pacific corals)	3	Animalia	[56]
*Pseudophryne* (frogs)	3	Animalia	[57]
*Steatocranus* (cichlid fish)	18	Animalia	[58] *
*Stylophora* (Red Sea coral)	Not specified	Animalia	[59] *
*Sus* (wild pigs)	4	Animalia	[60]
*Ursa* (bears)	6	Animalia	[61]
*Xiphophorus* (fishes)	5	Animalia	[62]
*Abies* (fir)	3	Plantae	[63] (in [64] *)
*Actinidia* (Kiwi)	9	Plantae	[21,65] ^2^
*Aesculus* (buckeye)	3	Plantae	[66] (in [64] *)
*Ajuga* (bugleherb) or *Amaranthus* (amaranths)	5	Plantae	[64] *
*Ambrosia* (ragweed)	3	Plantae	[64] *
*Amelanchier* (serviceberry)	5	Plantae	[67] (in [64] *)
*Aquilegia* (Columbines)	Not specified	Plantae	[68] *
*Arbutus* (madrones)	5	Plantae	[69]
*Arctostaphylos* (manzanita)	At least 3	Plantae	[70,71]
*Asclepias* (milkweed)	4	Plantae	[64] *
*Asplenium* (spleenworts)	16	Plantae	[72] (in [15] *)
*Betula* (birch)	One set of 4; one set of 6	Plantae	Gunnarsson in [4] *; [73,74] (in [64] *)
*Boechera* (rockcress)	58	Plantae	[75]; D. Bailey (in [15] *)
*Carex* (true sedges)	Three sets of 3; two sets of 4	Plantae	[76] *
*Castanea* (chestnut)	Not specified	Plantae	[77] *
*Ceanothus* (California lilac)	Not specified	Plantae	[78] *
*Cirsium* (plume thistle)	17	Plantae	[79] *
*Citrus*	8	Plantae	[80,81] ^2^
*Coprosma* (stinkwood)	6	Plantae	[82] *
*Cornus* (dogwood)	4	Plantae	[64] *
*Corybas* (helmet orchid)	3	Plantae	[83]
*Cyperus*	3	Plantae	[84] (in [64] *)
*Dichanthelium* (rosette grass)	Two sets of 3; one set of 4	Plantae	[64] *; [85] (in [64] *)
*Diospyros* (ebonies)	One set of 3, one set of 4	Plantae	[86] *
*Diplacus* (monkey flower)	5	Plantae	[78,87] *
*Drosera* (sundew)	4	Plantae	[88] (in [64] *); [89] (in [64] *)
*Dryopteris* (woodfern)	4	Plantae	[90] (in [64] *)
*Dubatia*	6	Plantae	[91]
*Elymus* (wildrye)	3	Plantae	[64] *
*Encelia* (brittlebush)	11	Plantae	[92] *
*Equisetum* (horsetail)	3	Plantae	[64] *
*Eschweilera*	3	Plantae	[93] *
*Espeletia* (frailejones)	3	Plantae	[94]
*Eucalyptus* (Green ashes)	4	Plantae	[95]
*Eucalyptus* (Boxes)	~10	Plantae	[96]
*Ficus* (figs)	13	Plantae	[97]
*Gentiana*	4	Plantae	[64] *
*Geum* (avens)	Not specified	Plantae	[5] (in [78] *)
*Gymnocarpium* (oak fern)	4	Plantae	[98] (in [64] *)
*Helianthus* (sunflower)	One set of 4; one set of 6	Plantae	[64,99,100,101,102,103] ^1^
*Hieracium* (hawkweed)	One set of 4; one set of 5	Plantae	[64] *
*Huperzia* (firmosses)	3	Plantae	[64] *
*Hypericum* (St. John’s wort)	3	Plantae	[64] *
*Iris* (California irises)	12	Plantae	[78,104] ^2^; [105] * (in [15] *)
*Juncus* (rushes)	7	Plantae	[106] *
*Juniperus* (junipers)	3	Plantae	[107,108] (in [64] *)
*Lantana*	6	Plantae	[109] *
*Lespedeza* (bush clovers)	5	Plantae	[64] *
*Ligularia* (leopard plants)	3	Plantae	[110]
*Lycopodiella* (bog clubmosses)	4	Plantae	[111] (in [64] *)
*Lycopus*	4	Plantae	[64] *
*Lysimachia*	3	Plantae	[64] *
*Melandrium*/*Silene* (campion)	Not specified	Plantae	[5] (in [78] *)
*Micromeria*	20	Plantae	[112] *
*Nothofagus* (southern beeches)	At least 3	Plantae	[5] (in [78] *); [113,114]
*Opuntia* (prickly pear cactus)	At least 16	Plantae	[115] *
*Phaseolus* (bean)	3	Plantae	[116] *
*Phlox*	3	Plantae	[117] (in [64]*)
*Picea* (spruces)	3	Plantae	[118]
*Pinus* (Southwestern pinyon pines)	4	Plantae	[119,120] *
*Platanthera* (butterfly orchids)	Two sets of 3	Plantae	[121] (in [64] *)
*Populus* (cottonwood)	Three sets of 3	Plantae	[122] (in [64] *); [123] (but see [23] *)
*Potamogeton* (pondweed)	19	Plantae	[78] * (in: [15] *); [124]
*Prosopis* (mesquite)	7	Plantae	[125,126] *
*Prunus* (plums)	18	Plantae	[127] *
*Pycnanthemum* (mountain mints)	3	Plantae	[64] *
*Quercus* (Chinese oaks)	4	Plantae	[128]
*Quercus* (Eastern white oaks)	14	Plantae	[78,129] *
*Quercus* (Southwestern white oaks)	16	Plantae	[78] *; R. Spellenberg (in: [15] *)
*Rosa* (rose)	3	Plantae	[130] (in [64] *)
*Rubus* (brambles)	3	Plantae	[64] *
*Salix* (willow)	Two sets of 3; one set of 6	Plantae	[85] (in [64] *); [131]
*Saxifraga* (saxifrages)	3	Plantae	Lloyd in [4] *
*Schiedea*	4	Plantae	[132]
*Scirpus* (club-rush)	3	Plantae	[64] *
*Senecio*	5	Plantae	[19] *
*Solidago* (goldenrods)	One set of 4; one set of 5	Plantae	[64] * (but see [133])
*Sphaeralcea* (globemallows)	Not specified	Plantae	[134]
*Stipa*	Two sets of 3	Plantae	[135,136]
*Symphonia*	3	Plantae	[137] *
*Symphyotrichum*	8	Plantae	[138] (in [64] *)
*Thalictrum* (meadow-rue)	3	Plantae	[64] *
*Tolumnia* (Dancing-lady orchid)	4	Plantae	[139] *
*Tragopogon* (salsifies)	5	Plantae	[140,141]
*Trillium*	One set of 3; one set of 4	Plantae	[64] *; [142] * (but see [143])
*Tripsacum* (gamagrass)	7	Plantae	[144] *
*Verbascum* (mullein)	4	Plantae	[64] *
*Verbena* (vervain)	4	Plantae	[64] *
*Viola*	One set of 4; one set of 5; one set of 7	Plantae	[64] *

* All cited studies used the term “syngameon”. ^1^ Only the first study uses the term “syngameon”. ^2^ Only the second study uses the term “syngameon”.

## Data Availability

Not applicable.

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
