# Peer review of "The Syngameon Enigma"

_plants, 2022, doi:10.3390/plants11070895_

Round 1
Reviewer 1 Report
Brief summary:
The submitted manuscript reviews the current knowledge on syngameons, complex hybridisation networks among three or more closely related species. The review introduces the subject, describing several interpretations of syngameons. It includes an exhaustive list of organisms known to engage in syngameons accompanied with a reference list. The review then describes several different hypotheses on and the mechanisms behind the formation and collapse of syngameons, illustrated by figures. The authors then review how syngameons are maintained over time, discussing the advantages and disadvantages of engaging in syngameons, the impacts of the directionality of geneflow and the number of partners. The authors then question why syngameons are so rare in the literature, suggesting that the previous limitations to detect them and the historical dismissal of hybrids as quirks of nature are behind their rarity in the literature. Current genomic approaches are contributing to an increase in the detection of syngameons. The article finishes with a discussion on the impact of the syngameon in the conservation of biodiversity, which is frequently species focused.
General comments:
First of all, I must compliment the authors on such an extensive review of syngameons. I have absolutely loved reading this review on such a fascinating topic. Although, several reviews on syngameons on specific examples exits, the question needed a review encompassing the whole field. The authors have included a wide range of examples, including both plants and animals. They have included a wide diversity of articles tackling syngameons from many angles: morphology, genetics, genomics, conceptual and philosophical, their ecological and evolutionary relevance, and the conservation implications. The articles span a wide temporal window, with a good balance of classic articles on the question and a fair share of very recent publications on the genomics of syngameons.
The text is clearly written and accessible to a wide audience, explaining the concepts in an amenable manner (but see specific comments). The figures explain clearly and aid the understanding of the different concepts discussed on the text (but see specific comments). The ideas are well developed, and each section finished with suggestions on specific topics that needs addressing. The appendix is helpful at explaining what the authors mean by the different terms. Perhaps "Glossary" would be a better term for the appendix.
My only suggestion would be to expand a little on the "ways going forward" in the conclusion (Lines 465 - 471). Do you have suggestion on how to address these questions?
Overall, I think this manuscript will constitute an excellent and thought-provoking review of the subject and contribute to advancement on the knowledge of syngameons.
Specific comments:
Lines 56-57: weird sentence construction, had to read it several times to grasp what was meant. Suggest rephrasing: Only recently have researchers begun to hypothesize how these rare complexes are able to overcome numerous reproductive barriers in the process of their formation.
Lines 128-136: This section is slightly awkward to read. Instead of juxtaposing the two scenarios continuously, I think it would be better two explain one scenario, and then the following. Might make it easier to follow.
Line 217: substitute "this hypothesis" with the actual hypothesis for ease of reading.
Lines 232-235: I somewhat disagree with this statement. There is evidence that congeneric species of Neotropical trees frequently live in the same habitats (Allie et al, 2015; Canon & Lerdau, 2019 and references therein). Although congenerics may segregate into microhabitats (Allie et al, 2015; Schmitt et al, 2021b), the geographic distance between microhabitats is smaller than their pollen and fruit dispersal capacity, as demonstrated by the increase haplotype sharing among congenerics when congenerics are sympatric (Caron et al, 2019).
Allié E, Pélissier R, Engel J et al. (2015) Pervasive local-scale tree-soil habitat association in a tropical forest community. PLoS ONE, 10, 1–16.
Caron H, Molino JF, Sabatier D et al. (2019) Chloroplast DNA variation in a hyperdiverse tropical tree community. Ecology and Evolution, 9, 4897–4905.
Schmitt S, Tysklind N, Derroire G, Heuertz M, Hérault B (2021) Topography shapes the local coexistence of tree species within species complexes of Neotropical forests. Oecologia. 196:389–398 doi.org/10.1007/s00442-021-04939-2
Line 245: "joint" and "hubs". What is meant by these should be explained here, or at least point to the Appendix. Suggest adding: … hybridize with the same SINGLE species.
Line 395: suggest remove "habitable".
On the other hand: To me this is non-academic English (or at least it needs a previous sentence starting with "on the one hand". I suggest changing to "Conversely" or "on the contrary".
Figures:
I am not colour blind, so I can't comment on whether the colours are visible to colourblind people, but I would recommend that the authors check if the colour choices are distinguishable by colourblind people.
Fig. 2: text size of number might a little small.
Fig. 6: I don’t see the point of the dotted line. It's more distractive than informative. Perhaps this graph could be visually improved (ggplots). Also, I would suggest adding major breakthroughs in the history of syngameons (e.g. description of syngameons, microsatellites, genomics).
Reviewer 2 Report
The authors provide a review of syngameons, how they form, how are they maintained over time, why they are rare, and discuss syngameons in conservation biology.
I have relatively few comments to the ms. This review is well-structured, well-written and gives an overview of the subject - but the flip side to that is that it is also relatively superficial and light-weight as it does not dig deeply into the cited literature or evaluate the quality of that research.
A few aspects should be clarified in the introduction. Do interspecific hybrids within a syngameon have to be fertile or semifertile, thereby allowing for introgression among species and homoploid hybrid speciation, or whether it is still a syngameon if the hybrids are sterile? How about diploid-polyploid systems with hybridisation? Another term that is sometimes encountered in the literature is coenospecies. The authors should mention this term and clarify the difference between syngameon and coenospecies.
Concerning the detection of past and ongoing events of introgression (lines 384-386) within a radiation, it might be added that this can be very difficult in cases where population sizes are large and, as a result of this, coalescence times long and gene tree stochasticity high. Simplified methods like the ABBA-BABA test / D statistic cannot in fact distinguish between ancestral population structure and actual introgression (e.g. doi:10.1093/molbev/msu269), while more parameter-rich methods will typically assume that population sizes are constant, which is an unrealistic assumption during a species radiation.
The main reason why syngameons are rare is because they are bound to be evolutionarily ephemeral. They can occur in the early phase of diversification of a lineage, but before the daughter species have become too different to hybridise. It would be useful if the authors could indicate a timescale for how long syngameonsd persist, to the extent that this information exists in the literature. My own (highly inofficial) rule of thumb is that homoploid speciation takes 1 Ma and sister species may exchange genes up to no more than 5 Ma after diversification, but the ability to form interspecific hybrids (and allopolyploids) may persist for up to 60 Ma in certain lineages, such as ferns (https://www-nature-com.ezproxy.uio.no/articles/518276c).
